# Nitrified Human Urine as a Sustainable and Socially Acceptable Fertilizer: An Analysis of Consumer Acceptance in Msunduzi, South Africa

**Benjamin C. Wilde** [1,*], **Eva Lieberherr** [2] 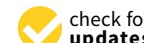, **Andrew E. Okem** [3] **and Johan Six** [1]

1   Sustainable Agroecosystems Group, ETH Zurich, Zurich 8092, Switzerland; jsix@ethz.ch
2   Natural Resource Policy Group, ETH Zurich, Zurich 8092, Switzerland; eva.lieberherr@usys.ethz.ch
3   School of Life Sciences, University of Kwazulu-Natal, KwaZulu-Natal 3629, South Africa; okem@ukzn.ac.za
*   Correspondence: benjamin.wilde@usys.ethz.ch; Tel.: +41-078-708-4584

**Abstract:** Sustainable smallholder farming is contingent on fertilizer access. Soils across Africa are typically nutrient deficient, a condition exacerbated by long-term nutrient mining. Nitrified urine fertilizer is a nutrient-rich and hygienically safe solution derived from human urine. It has the potential to provide a sustainable source of soil nutrients to low and middle-income countries struggling with food insecurity challenges. This study presents findings of a survey that assessed public acceptance within Msunduzi, Kwazulu-Natal, South Africa towards the use of nitrified urine fertilizer. Results indicate that in general attitudes were much more positive towards the use of nitrified urine fertilizer than raw urine as a soil amendment. Residents living within rural zones of the municipality (78.5%), as opposed to urban (65.7%) and peri-urban (65.2%), and younger individuals within the sampled population were found to be the most receptive to the use of nitrified urine fertilizer. Our findings also underscore the complex set of factors that shape attitudes towards a topic such as the use of human waste as a fertilizer, which are crucial in shaping the legitimacy of an emerging technology such as urine nitrification.

**Keywords:** sustainability; nitrified urine; consumer acceptance; fertilizer; Msunduzi; South Africa

## 1. Introduction

Smallholder agricultural production is ubiquitous across Africa. These systems serve as a means of increasing available supplies of food for both subsistence and to diversify revenue streams [1]. From rural to peri-urban and even into urban areas of human settlements on the continent, smallholder production is an important livelihood strategy and provides a key buffer to price volatility, long distances to markets, and low incomes. However, a major limiting factor for smallholder producers in Africa is a chronic lack of fertilizer. Soils across the region are typically nutrient deficient, a condition exacerbated by long-term nutrient mining [2]. This biophysical reality impedes the sustainability of smallholder farming as a livelihood strategy.

Recent advances in sanitation technology have the potential to change this. The Swiss Federal Institute of Aquatic Science and Technology (EAWAG) recently developed a method and technology to separate and process human urine into a concentrated, nutrient-rich solution as part of a distributed sanitation solution for peri-urban communities in South Africa [3]. Just as with raw human urine, which has been used as a source of soil nutrients for millennia in countries across Asia [4], nitrified urine is an effective fertilizer. Recent greenhouse trials show that the uptake of nitrogen and phosphorus from nitrified urine fertilizer (NUF) by plants is similar to commonly used chemical fertilizers [5]. Additionally, this process eliminates pathogen risks, reduces excreted pharmaceuticals to the limit of

detection, and stabilizes the nitrogen within the solution as ammonium nitrate [6,7]. Thus, sanitation systems that incorporate the ability to produce nitrified urine can yield a locally available and hygienically safe organic fertilizer.

The technical challenges involved in the production of NUF are largely solved; reactors have been in operation in both Europe and South Africa for several years. An initial biophysical trial [5] indicated that NUF is as available for plant uptake as are commonly utilized industrially produced fertilizers. Given this, it seems clear that NUF can serve as a central component for a socio-technical transition to help countries in Africa achieve more sustainable agroecosystems. However, socio-technical transitions typically require that new and innovative solutions must usually "overcome the rigidities and path-dependencies of already existing, highly institutionalized system structures" [8] (pp. 772–773). At the moment, a central obstacle, or rigidity, towards the incorporation of recycled human nutrients to improve agricultural productivity is negative public attitude. A study conducted in Nigeria and Ghana, for example, found that at least half the respondents objected to the idea of using raw urine as fertilizer, primarily because of cultural and religious concerns [9]. A separate study, also in West Africa, found that offensive odor (52%) and health concerns (21%) were the principal sources of negative perceptions towards the use of human waste as fertilizer [10]. South African studies, meanwhile, have found that the major concern of utilizing urine for agricultural purposes is related to potential negative health impacts [11,12]. Negative attitudes such as these are a major reason why the use of human urine, despite its potential to improve agricultural output, is largely ignored as a viable solution to the food insecurity challenge facing Africa. We thus address the following questions: Are South Africans more accepting of treated than raw urine? Which factors affect South African acceptance of raw versus nitrified urine?

Terms such as acceptance, acceptability, support, and adoption are commonly used in the technology acceptance literature, thus making a clear articulation of acceptance in this context necessary. For example, Batel et al., (2013) distinguish between acceptance and support, with the former implying a level of passive approval, while the latter term expresses active engagement for a specific technology or innovation [13]. Taking this concept of acceptance further, Huijts et al, (2012) delineate consumer and citizen acceptance, with the primary difference being the agency or freedom of choice of the individual to choose to be in contact with a given technology or innovation inherent in consumer acceptance [14]. In this paper, we focus on consumer acceptance to capture attitudes towards the use of NUF as a fertilizer. Numerous studies have found that factors such as religion, culture, or socio-economic status influence attitudes towards nutrient recycling [10,15]. We build on this literature and systematically identify which specific factors- hunger, race, age, religion, income, gender, education, and ward type- were significant in influencing attitudes towards NUF.

Our analysis is based on a survey campaign of households in Msunduzi municipality of South African. We first quantify the rates of food security, the relationship between poverty and food access, and the role of small-scale agriculture within the food landscape of Msunduzi to understand possible factors affecting the acceptance of NUF. We then assess acceptance by first quantifying the change in acceptance towards urine recycling through the processing of raw urine to nitrified urine concentrate. Second, we address acceptance by analysing the respondents' willingness to purchase food grown with NUF and differentiating between rural and urban areas and the sources of continued concern against NUF.

## 2. Materials and Methods

To assess consumer acceptance of NUF in Africa, we focused on Msunduzi, South Africa, a region that is considered representative of how cities across the continent will likely develop in the coming years [16]. Like most urban centers across lower and middle-income countries, this region includes burgeoning urban areas, such as Pietermaritzburg, which is the largest city within Msunduzi, with a population growth rate of 1.12 % [17]. Consequently, municipal actors face the challenge of providing

adequate services (sanitation, housing, power) to its growing population. This process of urbanization has created a distinct rural-urban spectrum, which is rapidly changing traditional land use patterns.

There are approximately 163,993 households in Msunduzi. According to municipal categories of race/ethnicity, 81.1% of the population is Black African, 2.9 % colored, 9.8% Indian/Asian, 6% White, and 0.3% other. The majority of the population speaks either isiZulu or English. The general unemployment rate is 33% and youth unemployment is 43.1% [17]. Furthermore, the region is highly integrated into the global food system, with many of the largest grocery store chains well represented in the city. Despite this, food insecurity is a major challenge facing the population of the region, with 60% of households classified as being food insecure [16].

For this study, we first delineated wards, or geopolitical subdivisions within the municipality, into rural, peri-urban, or urban typologies. This classification was done by analyzing the population density, the percentage of households connected to the municipal sanitation system, and access to municipal garbage services across all the wards of Msunduzi. Once divided, a stratified cluster sampling approach was adopted to randomly identify survey zones (Figure 1).

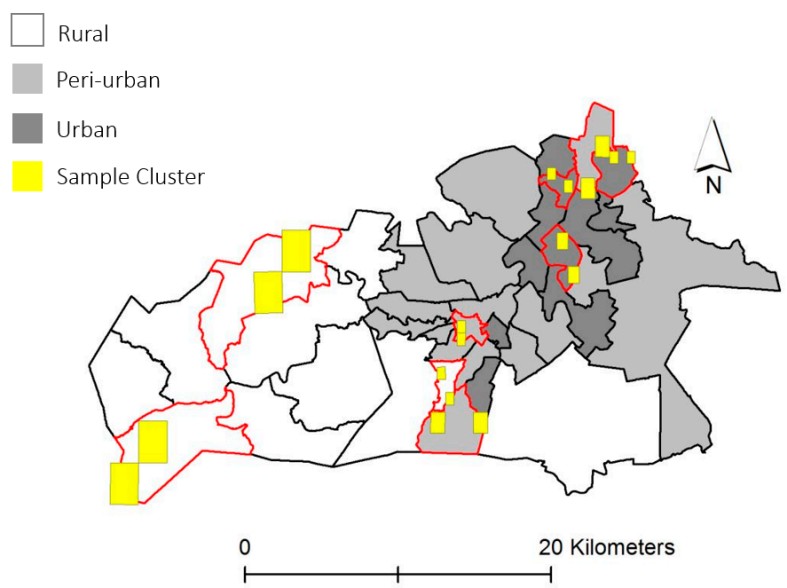

**Figure 1.** Msunduzi municipality by ward type. Yellow rectangles indicate the randomly selected survey zones.

A questionnaire was then developed to understand the food security situation of Msunduzi, as well as to identify specific factors that impact residents' attitudes towards food grown with NUF. Relative food insecurity was determined by asking how often the respondent felt it was difficult to provide enough food for him/herself and the household. The respondents were given four possible options: always, most of the time, rarely, or never. A household that responded always or most of the time was considered food insecure, any household that answered either rarely or never was considered food secure. Participation in smallholder agriculture was quantified by asking respondents whether they engage in agricultural production, if so what proportion of their food is sourced from this activity, and whether they grow food for subsistence or income generation. To assess the impact of the treatment process of NUF on public acceptance, a logistic regression model was constructed to identify factors driving this change in opinion. To construct this model, the score difference between respondent answers from questions focused on raw urine and NUF was calculated. These scores were then grouped into two categories: those respondents whose answers did not change (0) and those whose answers changed significantly (>75). This categorical dependent variable was then evaluated against eight demographic predictors (hunger, race, income, religion, age, gender, education, and ward). Willingness to consume food grown with NUF was identified by asking respondents a

four-point scale with yes, most likely yes, most likely no, and no being the available answer options. Pearson's $\chi^2$ tests were used to test for independence between pertinent variables.

Working in collaboration with researchers from the University of Kwazulu-Natal, 11 local bachelor students were recruited to conduct the fieldwork, which occurred over the course of 10 days in October 2017. Data were collected with mobile phones equipped with the software platform "Mobenzi Researcher" and automatically uploaded to a central web console. A sample size of 392 household questionnaires with a response rate of 86 % was obtained during this period. Coding and data analysis was performed using Statistical Package for the Social Sciences (SPSS) version 24 as well as Rstudio1.

## 3. Results

392 household questionnaires were collected from rural, peri-urban, and urban wards across Msunduzi during the survey campaign (Figure 2). Table 1 provides the demographic characteristics of the sample population. Below we present the results on acceptance and for the influencing factors.

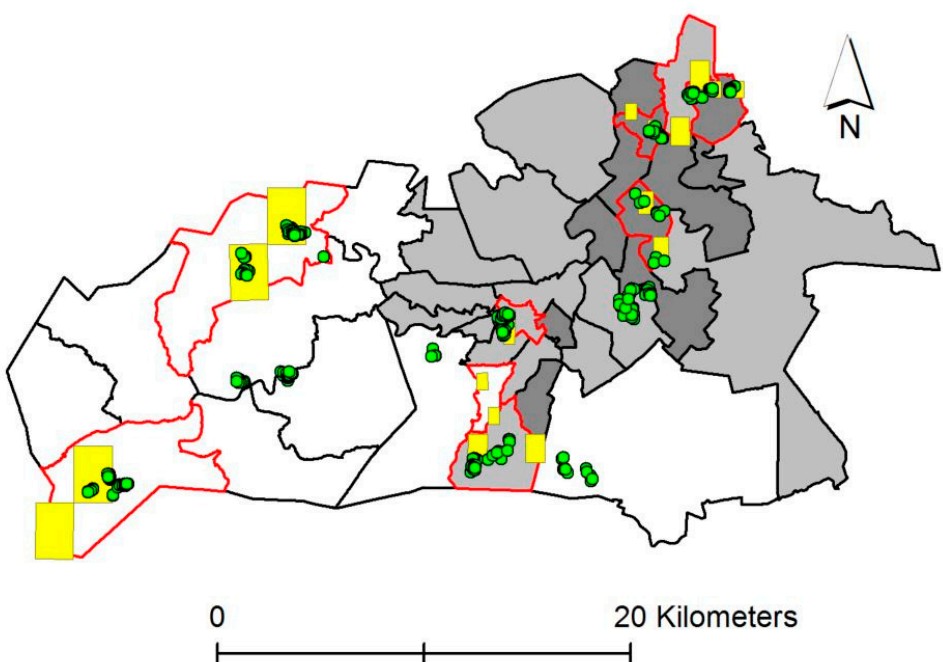

**Figure 2.** Map showing the location of conducted surveys.

### 3.1. Food Insecurity

We found that roughly half (47.6%) of the households in Msunduzi can be considered food insecure. When analyzed by ward, the peri-urban communities suffer the highest level of food insecurity (Figure 3). Interestingly, the urban wards, despite having a large number of food retailers and being highly integrated in the global food network, still suffer from very high levels of food insecurity (38.1%). A $\chi^2$ test of independence conducted between ward and food insecurity was significant ($\chi^2$ (2, N = 382) = 6.843, $p$ = 0.033), indicating that ward does influence rates of food security within Msunduzi. Based on an odds ratio analysis, we found that the likelihood of being food secure in an urban ward of the municipality is 1.95 times higher than compared to a peri-urban ward.

**Table 1.** Descriptive statistics of the sampled population.

| Gender | Number of Responses | Response Percentage |
|---|---|---|
| Male | 157 | 40.0 |
| Female | 235 | 59.9 |
| **RACE/ETHNICITY** | | |
| Black/African | 338 | 86.2 |
| Indian | 30 | 7.6 |
| White | 19 | 4.8 |
| Colored | 5 | 5 |
| **AGE** | | |
| 16–30 | 129 | 33.0 |
| 31–45 | 111 | 28.3 |
| 46–60 | 78 | 20.0 |
| 61 + | 58 | 14.8 |
| No Response | 14 | 3.5 |
| **INCOME BRACKET** | | |
| 0–1500 | 187 | 48.1 |
| 1501–3500 | 51 | 13.1 |
| 3501–5000 | 21 | 5.4 |
| 5001–10000 | 7 | 1.8 |
| 1001–15000 | 11 | 2.8 |
| >15000 | 13 | 3.3 |
| No Response | 98 | 25.2 |
| **EDUCATION** | | |
| <Primary School | 36 | 9.2 |
| Completed Primary | 98 | 25.0 |
| High School | 180 | 46.0 |
| University | 46 | 11.7 |
| No Response | 31 | 7.9 |
| **RELIGION** | | |
| Christianity | 300 | 76.7 |
| Islam | 22 | 5.6 |
| Hinduism | 13 | 3.3 |
| Judaism | 0 | 0 |
| None | 14 | 3.5 |
| Other | 31 | 7.9 |
| No Response | 11 | 2.8 |

Additionally, there is a significant relationship ($\chi^2$ (3, N = 287) = 9.215, *p* = 0.027) between income and food insecurity. Households surveyed that earn no more than R 1500/month (equals approximately 110 USD/month), which comprised 48.1% of the total sampled population, were more likely to answer that they either always or mostly find it difficult to secure enough food for themselves and their families (Figure 4). This intuitively makes sense and does indicate that hunger in Msunduzi is inextricably linked with poverty.

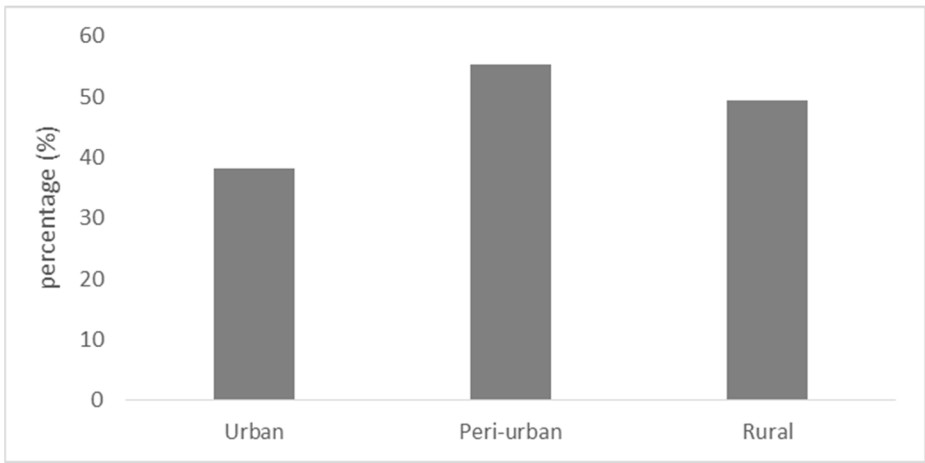

**Figure 3.** Proportion of population that is food insecure by ward type.

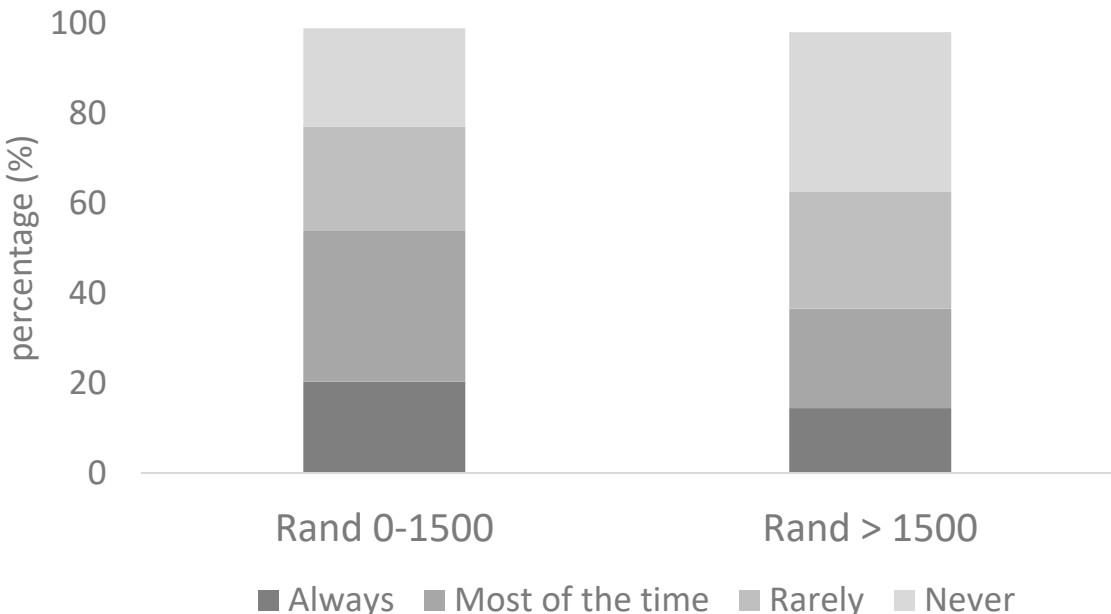

**Figure 4.** Role of income (South African Rand) in influencing food security.

### 3.2. Small-Scale Agriculture

Although it may be the case that smallholder production is not as widespread as in the past or in comparison to other African countries, we found that it is still an important factor for many households living in Msunduzi, with 54.7% of all respondents stating that they engage in some sort of agricultural activity (Table 2).

In rural areas, 64.4% of respondents answered that they engage in some type of agricultural activity. Respondents in both the peri-urban (48.2%) and urban (51.5%) zones also stated that they engage in some form of agriculture. By far the major reason given for engaging in this activity is to supplement diet (71%). A quarter of respondents claim to grow food both for subsistence as well as to sell, and only 2.3% stated that they grow solely to supplement income. A $\chi^2$ test of independence found no relationship between a household's relative food security and whether or not they engage in agricultural activity.

**Table 2.** Proportion of residents currently engaged in some form of agricultural activity.

| Engaged in Agriculture | Number of Responses | Response Percentage |
|---|---|---|
| Yes | 214 | 54.7 |
| No | 174 | 44.5 |
| No response | 3 | 0.7 |
| Purpose of Agriculture | | |
| Subsistence | 152 | 71.0 |
| Income | 5 | 2.3 |
| Both | 54 | 25.2 |
| Other | 3 | 1.5 |

*3.3. Acceptance*

3.3.1. Attitude Toward NUF vs Raw Urine

Our findings suggest that acceptance changes drastically when respondents consider NUF rather than raw urine as a proposed soil amendment. The histogram below (Figure 5) displays the results from two related statements to which respondents were asked to indicate their level of agreement versus disagreement, one regarding the use of treated urine (Question 1), and the other focused on the use of raw urine (Question 2).

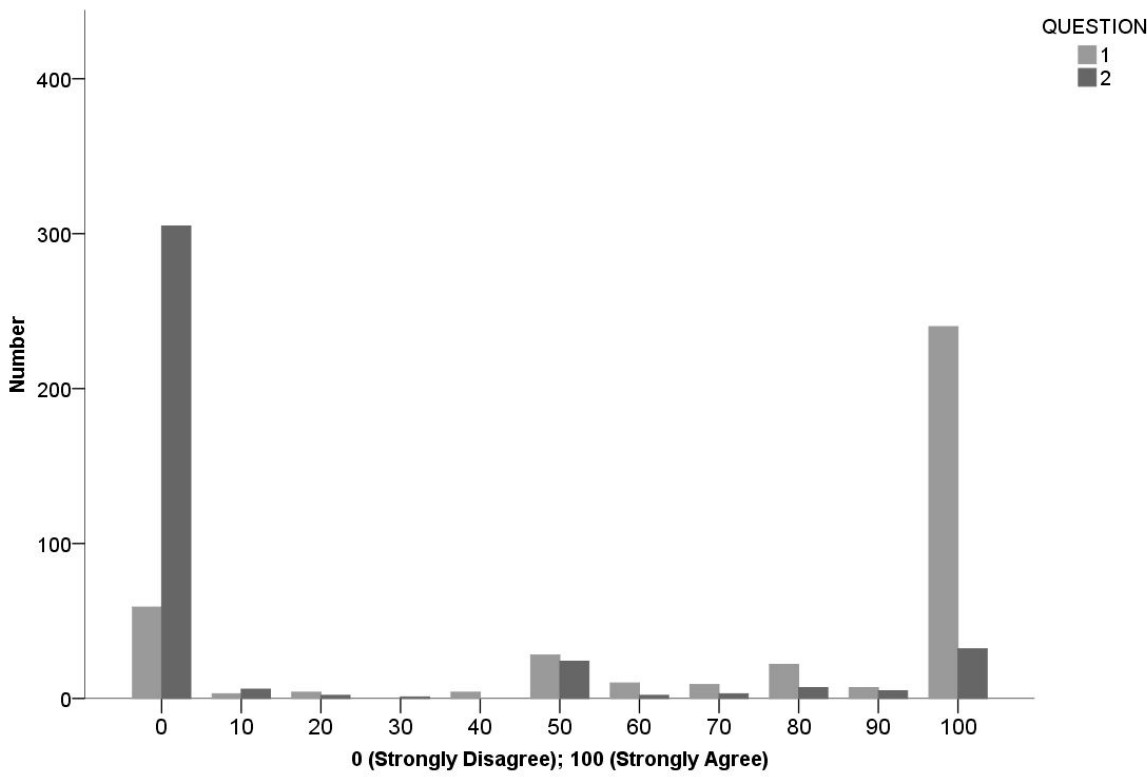

**Figure 5.** Responses to questions related to the acceptance of raw urine (2) compared to treated human urine (1) as a potential source of fertilizer for food crops.

Question 1: Recycled human urine, if treated properly to ensure it is safe, would make a suitable fertilizer.

Question 2: The use of untreated urine, because of its high nutrient content, can serve as a fertilizer for food crops.

The mean value respondents gave for question 1, which asked for an opinion on the use of treated urine, was 75.8, while the mean value of respondents towards untreated urine was 15.2. These results suggest that public attitudes towards the utilization of NUF as a fertilizer for food crops are much more favorable than attitudes versus the use of raw urine. Furthermore, a logistic regression model was constructed (Table 3) to identify factors driving this change in opinion.

**Table 3.** Results from logistic regression to identify significant predictors. * $p < 0.10$, ** $p < 0.05$.

| Change in Opinion | B (SE) | Lower | 95% CI FOR OD | Upper |
|:---:|:---:|:---:|:---:|:---:|
| Intercept | −0.50 (0.48) | 0.15 | 0.60 | 2.42 |
| HUNGER 2 | 0.11 (0.78) | 0.51 | 1.12 | 2.45 |
| HUNGER 3 | −0.23 (0.57) | 0.35 | 0.80 | 1.80 |
| HUNGER 4 | 0.44 (0.27) | 0.71 | 1.55 | 3.46 |
| RACE 2 | 1.60 (0.05) | 1.07 | 4.95 | 28.71 |
| RACE 3 | 0.54 (0.38) | 0.51 | 1.71 | 5.79 |
| INCOME 2 | −0.08 (0.78) | 0.50 | 1.00 | 1.70 |
| RELIGION 2 | −0.87 (0.31) | 0.07 | 0.42 | 2.09 |
| RELIGION 3 | −1.08 (0.27) | 0.04 | 0.34 | 2.24 |
| RELIGION 4 | −0.42 (0.34) | 0.27 | 0.66 | 1.57 |
| AGE 2 | −0.79 (0.02) * | 0.23 | 0.46 | 0.90 |
| AGE 3 | −0.97 (0.01) * | 0.17 | 0.38 | 0.82 |
| AGE 4 | −0.54 (0.25) | 0.23 | 0.58 | 1.50 |
| GENDER 2 | 0.19 (0.50) | 0.70 | 1.21 | 2.10 |
| EDUCATION 2 | 0.16 (0.78) | 0.41 | 1.17 | 3.41 |
| EDUCATION 3 | 0.54 (0.30) | 0.62 | 1.72 | 4.94 |
| EDUCATION 4 | −0.19 (0.78) | 0.21 | 0.82 | 3.28 |
| WARD 2 | 0.33 (0.37) | 0.69 | 1.38 | 2.85 |
| WARD 3 | 1.15 (0.005) ** | 1.44 | 3.15 | 7.13 |

Note. R2 0.08 (Hosmer-Lemeshow), 0.10 (Cox-Snell), 0.14 (Nagelkerke). $\chi^2$ (1) = 15.96, $p = 0.006$

Within the eight predictors, the youngest age cohort sampled (16–30) and respondents living in the rural wards had difference scores that were significant. The proportion of respondents from each ward type that expressed positive attitudes towards the purchase or consumption of food grown with NUF was also assessed (Figure 6).

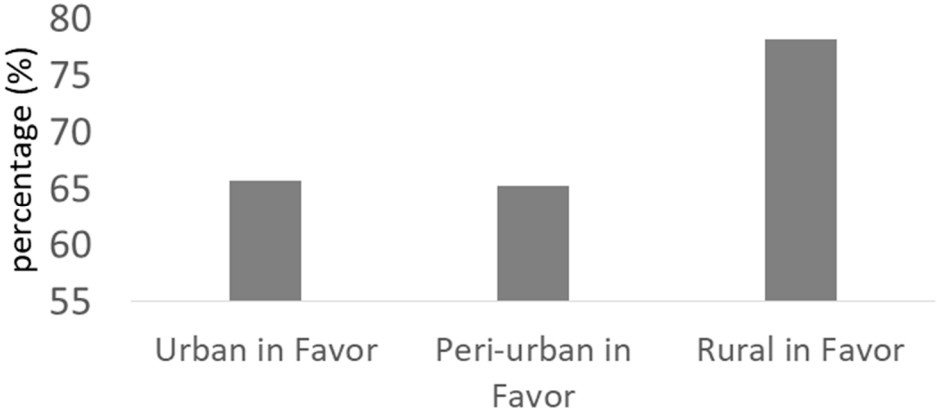

**Figure 6.** Percentage of respondents living within each ward type with positive attitudes towards purchase of food grown with NUF.

However, many respondents continued to express reservations towards the use of NUF as a fertilizer. Although multiple reasons were cited for this, a lack of trust in the processing of urine to NUF and thus a continued concern regarding the health implications was the most common source of this reservation (Figure 7).

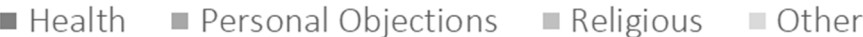

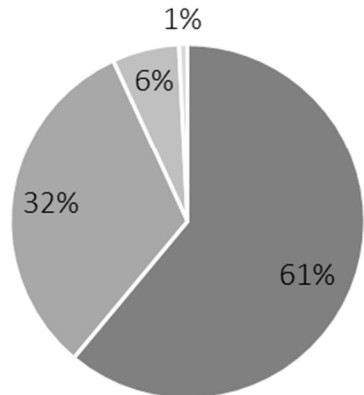

**Figure 7.** Sources of continued concern regarding the use of NUF as a soil amendment.

### 3.3.2. Willingness to Purchase Food Grown with NUF

Respondents were also asked whether they would be willing to purchase food if they knew it was grown with NUF. Table 4 displays the results of a series of $\chi^2$ tests that assessed the relationship between willingness to purchase food grown with NUF and the following independent variables: ward, sex, income, education, race, and engagement with agriculture. The only statistically significant results were between willingness to purchase NUF grown food and ward type ($\chi^2$ (6, N = 391) = 16.513, $p$ = 0.011) and willingness to purchase NUF food and race ($\chi^2$ (3, N = 392) = 9.048, $p$ = 0.029). Residents within the rural areas of Msunduzi showed the highest (78.2%) willingness to purchase food grown with NUF (Figure 7).

**Table 4.** Willingness to purchase food grown with NUF by demographic predictors.

|  | Yes | Most Likely Yes | Most Likely No | No | Total | $\chi^2$ |
|---|---|---|---|---|---|---|
| WARD |  |  |  |  |  | 0.011 |
| Urban | 45 (42.9%) | 24(22.9%) | 6 (5.7%) | 30 (28.6%) | 105 |  |
| Peri-Urban | 70 (41.9%) | 40 (24.1%) | 19 (11.4%) | 39 (23.4%) | 167 |  |
| Rural | 74 (62.2%) | 19 (16.0%) | 7 (5.9%) | 19 (16%) | 119 |  |
| GENDER |  |  |  |  |  | 0.827 |
| Male | 76 (49.7) | 39 (23.4) | 12 (7.8) | 31 (20.3) | 153 |  |
| Female | 107 (46.5) | 48 (20.9) | 20 (8.7) | 55 (23.9) | 230 |  |
| INCOME |  |  |  |  |  | 0.871 |
| 0–1500 | 81 (45.8) | 40 (22.6) | 17 (9.6) | 39 (22.0) | 177 |  |
| >1500 | 131 (47.3) | 63 (22.7) | 25 (9) | 58 (20.9) | 100 |  |
| EDUCATION |  |  |  |  |  | 0.544 |
| Did not finish primary school | 12 (36.4) | 10 (30.3) | 4 (12.1) | 7 (21.2) | 33 |  |
| Completed primary school | 49 (50.0) | 17 (17.3) | 7 (7.1) | 25 (25.5) | 98 |  |
| Completed high school | 88 (49.4) | 38 (21.3) | 15 (8.4) | 37 (20.8) | 178 |  |
| Completed university | 18 (47.3) | 14 (31.8) | 5 (11.4) | 7 (15.9) | 44 |  |
| RACE |  |  |  |  |  | 0.029 |
| Black (African) | 171 (50.7) | 65 (19.3) | 29 (8.6) | 72 (21.4) | 317 |  |
| Other (Indian, White, Colored) | 18 (32.7) | 18 (32.7) | NA | 16 (29.1) | 52 |  |
| AGRICULTURE |  |  |  |  |  | 0.065 |
| Engaged | 109 (50.7) | 51 (23.7) | 12 (5.6) | 43 (20.0) | 215 |  |
| Not Engaged | 80 (45.2) | 32 (18.1) | 20 (11.3) | 45 (25.4) | 177 |  |

## 4. Discussion

A key finding of this research is that overall attitudes within Msunduzi towards the use of NUF are much more positive than towards the use of raw urine as a fertilizer. This is in line with a previous study in South Africa, which found that negative attitudes towards nutrient recycling are shaped primarily by public health concerns [12]. We find that if these concerns are alleviated, it results in a shift towards greater acceptance of nutrient recycling from urine. As the data show, respondents answered much more favorably to the idea of utilization of NUF than towards raw urine as a fertilizer. However, the data also indicate that a lack of trust negatively affects acceptance. For example, 61% of respondents who stated that they would oppose the use of NUF as a fertilizer cited a continued fear of possible health problems as the primary reason. Additionally, when offered the opportunity to specifically state the cause of this fear, many respondents indicated that they do not trust the process involved in the conversion of urine into NUF.

When analyzed more closely, the data showed that two predictors most influenced respondent attitudes towards the use of NUF as a fertilizer: age and ward type. The logistic regression model shows that younger respondents (16–30) and citizens living in the rural areas of the municipality displayed the largest magnitude of attitude change between the use of raw and treated urine for fertilizer use. Based on these results, we infer that the younger age cohorts are more open and trusting of the nitrification process employed to treat the urine than older respondents. Similarly, a study focused on the acceptance of urine diversion dry toilets in eThekwini municipality also found that older residents were less receptive to adopting the new technology [18]. Additionally, residents within the rural areas of the municipality also displayed a larger change in attitude due to the treatment process than their peri-urban or urban counterparts. This, in combination with chi-square results showing rural residents being more willing to consume food grown with NUF, indicates a higher level of acceptance for this technology in rural compared to urban and peri-urban zones of the study area.

Research in other parts of the continent indicate that cultural norms and religion serve as important drivers of acceptance of human urine as a fertilizer, which was not the case in our study. For example, a study in Nigeria and Ghana found consumers had specific religious (38%) and cultural objections (26%) to the use of human urine as a fertilizer [9]. In contrast, our study found only 6 % of residents objected to NUF for religious reasons. This is also similar to the percentage of residents (6 %) within the neighboring municipality of Ethekwini that objected to raw urine as fertilizer for religious reasons [12]. Our study and that of Okem [12] were conducted in adjacent municipalities, both of which are located within Kwazulu-Natal, and are comprised of populations with very similar demographic makeups. We can thus infer that for respondents whose attitudes are based on religion, the treatment process has no effect. This supports assertions that behavior towards sanitation and excrement is a complex interaction between individual and societal norms and is often not based on scientific logic or knowledge [19,20].

Our results also indicate that lack of trust in the technology (NUF) is an issue. Many respondents indicated that they were suspicious of the treatment process. Phrases such as "it's new and no proof", "I want to see the process of the urine being cleaned", "assurance of proper cleaning" were common when respondents were given an opportunity to explain their reluctance in their own words. These attitudes are not surprising. Trust in new technologies has been shown to take time [21]. Additionally, issues of trust and fairness have all been shown to influence public attitudes [14,22]. In South Africa, the provision of sanitation services makes this quite clear. Under apartheid, decades of preferential municipal spending on public services led to widespread distrust and perceived unfairness between citizens and the state [23]. This could perhaps help to explain why age was a significant predictor of acceptance towards NUF; the 16–31 cohort did not experience apartheid directly, and are less influenced by its legacies.

Given our results, it seems clear that the 'innovation journey' [24] of NUF is uncertain. According to Deuten et al. [25] the successful incorporation of a new technology within a given society requires that it align closely within three environments: the business environment, regulatory environment, and the

wider society. The degree to which an innovation can accomplish this alignment is dependent on the degree of cultural legitimacy it accrues [26] and is of particular importance in the early phases of the innovation journey. As our results show, this will depend in large part on the logic that formed public attitudes towards nutrient recycling. In regions where attitudes towards nutrient recycling are shaped primarily by public health concerns [27,28], it is likely that sanitation technology can accrue higher levels of legitimacy than in regions with attitudes shaped by other factors. Although regulatory bodies in countries such as Switzerland [29] have recently amended existing laws to allow the use of NUF as a fertilizer for food crops, the use of human waste as a fertilizer is currently banned in South Africa. Whether or not other regulatory agencies adopt similar regulatory changes is difficult to predict.

Given this uncertainty, we consider Fuenfschilling and Truffer's [8] suggestion that the successful dissemination of radical innovations requires the creation of a technical niche critical in the context of human based fertilizer use. This protected space allows for "the maturation of new technologies and alignment with a suitable institutional context." Going further, the transition literature [30,31] suggests this can be done through the provision of subsidies and research grants, or the creation of experimental implementation projects [32]. In our opinion, the widespread adoption and scaling up of emerging technologies such as urine nitrification that can improve the sanitation and food security landscape in low and middle-income countries will require the creation of these types of platforms.

Finally, to contextualize the relevance of our results, we have embedded our research in the general state of the food system within Msunduzi. Our findings on acceptance are highly relevant given this study area, as we found that poverty is a major driver of food insecurity in Msunduzi municipality. A $\chi^2$ test found a significant relationship between income and food insecurity ($p = 0.027$), indicating that households earning no more than R 1500/month (= 113 USD/month) suffer from very high rates of food insecurity. This income cohort comprises 48% of the total population of Msunduzi, indicating the widespread nature of this challenge. Furthermore, despite the large numbers of food retailers in the urban zones, high rates of food insecurity were identified in every area of the municipality. This reinforces the link between poverty and food insecurity within the study area and makes the use of human urine as a fertilizer more pertinent.

Beyond food insecurity, the role of smallholder agriculture also makes acceptance of urine fertilizer salient. While it may be true that this activity plays a more prominent role in other African countries [16], our study found that smallholder production continues to serve as an important livelihood strategy in Msunduzi; 54.7% of all respondents indicated that they engage in some form of agricultural production. Of those, the majority do so primarily for subsistence purposes. Although the rural areas are the most heavily invested in agriculture, the respondents in peri-urban and urban zones indicated a high level of involvement in this activity as well. This finding was unexpected and warrants further study. Interestingly, no significance was found between engagement with agriculture and attitudes towards the use of NUF.

## 5. Conclusions

We contribute to a growing body of evidence that suggests that NUF production can play a significant role in the challenge of increasing soil nutrient access for farmers in areas such as South Africa. Our findings indicate that consumer attitudes towards the use of recycled human waste can be positively shifted through the processing of raw waste such as urine into a treated product such as NUF. Specifically, we found that residents in rural wards and younger residents were the most open to the use of treated urine as fertilizer. However, we also conclude that the impact on existing attitudes is highly dependent on the societal norms and underlying logic upon which those attitudes were shaped. This underscores the complex dynamics involved in promoting a shift towards the wide scale adoption of nutrient recycling technologies and the critical importance in understanding the local context.

**Author Contributions:** This research was conducted as a collaborative effort between ETH Zurich (Sustainable Agro-ecosystems and Natural Resource Policy), and the University of Kwazulu-Natal. B.C.W., with the support of E.L., J.S., and A.E.O. conceptualized the research questions and designed the questionnaire.

B.C.W. and A.E.O. supervised data collection. Formal analysis was conducted by B.C.W. with the supervision of E.L. and J.S. Original draft preparation was conducted by B.C.W., and review and editing done by all authors. Project administration was overseen by J.S. and funding acquisition was obtained by J.S. and B.C.W.

**Funding:** This research was funded by Stiftung Mercator Schweiz as part of a collaboration between the World Food System Center and Mercator Research Program, call 5.

**Acknowledgments:** The authors would like to acknowledge the work and support of the enumerators from the University of KwaZulu-Natal who conducted the field campaign. We would also like to thank the respondents who gladly gave their time to participate in this study. Finally, we would like to thank the World food System Center and Stiftung Mercator Schweiz for funding this research through the Mercator Foundation.

**Conflicts of Interest:** The authors declare no conflict of interest. The founding sponsors had no role in the design of the study; in the collection, analyses, or interpretation of data; in the writing of the manuscript, and in the decision to publish the results.

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
