# Peer review of "Nitrified Human Urine as a Sustainable and Socially Acceptable Fertilizer: An Analysis of Consumer Acceptance in Msunduzi, South Africa"

_sustainability, doi:10.3390/su11092456_

Reviewer 1 Report

This paper used surveys to examine attitudes towards the use of nitrified human urine as a fertilizer. Although the survey was relatively shallow in its exploration of the topic (only two questions) the results are clear, methods sound, and the paper very well written.  I think the tables could be presented more clearly to help the reader more easily digest the information, but other than that have no real comments for this rather straightforward paper. 

Author Response

Point 1: I think the tables could be presented more clearly to help the reader more easily digest the information, but other than that have no real comments for this rather straightforward paper.

Response 1: The authors amended the tables to reflect the specific changes requested by the reviewers. 

Reviewer 2 Report

A well written paper on an important topic. Generally, the results are well presented, however, there are some minor comments.

I would like to see more comparable studies in the discussion section. For example the lack of urine diversion toilets (UD) in South Africa have an impact on your study and the resident awareness about the use of urine for agricultural purpose?

Line 284 Any other countries than Switzerland that have allowed the use of NUF as fertilizer for food crops?

Please check and add the appropriate references:

https://doi.org/10.1016/j.jclepro.2014.01.070 

https://doi.org/10.2166/wst.2003.0015      

https://doi.org/10.3390/proceedings2110606

Moreover, the manuscript needs some serious editing work before published (for example the wrong position of figures caption)    

Author Response

Point 1: 

I would like to see more comparable studies in the discussion section. For example the lack of urine diversion toilets (UD) in South Africa have an impact on your study and the resident awareness about the use of urine for agricultural purpose?

Response 1:

The suggested literature was incorporated into the discussion section. Urine Diversion Dry toilets are not widely utilized in the study region. Urban residents in Msunduzi are typically connected to a municipal, water born sanitation system. Rural residents within the study site almost all utilize a Ventilated Improved Pit Latrine (VIP). Given this, there is a lack of awareness on the part of residents in the area towards the use of urine as a fertilizer. Even in eTheKwini, a municipality adjacent to the study site which has explored the use of UDDT technology, a study by Okem (2013) found a lack of experience and awareness of urine fertilizer. However the authors do not consider this lack of awareness to impact the validity of the paper’s conclusions.

Point 2: 

Line 284 Any other countries than Switzerland that have allowed the use of NUF as fertilizer for food crops?

Response 2: 

The authors contacted engineers currently employed at VUNA, the Swiss company producing NUF, and were informed that Switzerland is currently the only country that has approved the use of NUF as a fertilizer for food crops.

Point 3: 

Moreover, the manuscript needs some serious editing work before published (for example the wrong position of figures caption)

Response 3:

The manuscript has been edited to reflect the suggested changes. 

Reviewer 3 Report

General comments

The paper deals with the use of nitrified human urine as a fertilizer. The topic is of interst and methodology is correct. Paper presentation should be improved.

Specific comments

- please check English language recurring to a mother tongue sientific editor;

- figure captions should be insderted under the figures and not over the figures

- figure 4, please clear the meaning of R (i think it is the currency symbol, but it that case it is preferred the complete name);

- figure 5, please substitute "count" with "number"

- Figures 3,4 and 6: you should write in the y axis "Percentage (%)"

- line 327, please delete "Supplementary Materials:"

Author Response

Point 1:

please check English language recurring to a mother tongue sientific editor

Response 1:

There is inconsistency between the reviewers regarding the quality of the writing. Reviewer 1 stated that the paper is “very well written.” Reviewer 2 stated that the paper is a “well written paper on an important topic.” However reviewer 3 stated, “please check English language recurring to a mother tongue sientific editor;” The manuscript authors assume that reviewer 3 was suggesting that the paper be referred to a native English speaker. The authors request a more definitive consensus regarding the quality of the writing before referring the manuscript to another native English speaker for revision.

Point 2:

figure captions should be insderted under the figures and not over the figures

Response 2:

The figure captions have been changed accordingly.

Point 3:

figure 4, please clear the meaning of R (i think it is the currency symbol, but it that case it is preferred the complete name)

Reponse 3:

The figure has been amended accordingly.

Point 4: 

figure 5, please substitute "count" with "number"

Response 4:

Count was substituted with number.

Point 5:

Figures 3,4 and 6: you should write in the y axis "Percentage (%)"

Response 5:

The y axis was corrected.

Point 6:

line 327, please delete "Supplementary Materials:"

Response 6:

Supplementary Materials was deleted.

Round  2

Reviewer 3 Report

All the suggested changes have been performed. Paper can be accepted.